# *Origanum syriacum* Phytochemistry and Pharmacological Properties: A Comprehensive Review

**DOI:** 10.3390/molecules27134272

**Published:** 2022-07-02

**Authors:** Joelle Mesmar, Rola Abdallah, Adnan Badran, Marc Maresca, Elias Baydoun

**Affiliations:** 1Department of Biology, American University of Beirut, Beirut P.O. Box 11-0236, Lebanon; jm104@aub.edu.lb (J.M.); rha62@mail.aub.edu (R.A.); 2Department of Nutrition, University of Petra, Amman 961343, Jordan; abadran@uop.edu.jo; 3Aix-Marseille Univ, CNRS, Centrale Marseille, iSm2, 13013 Marseille, France

**Keywords:** *Origanum syriacum*, Lamiaceae, herbal medicine, phytochemistry, pharmacology, antioxidant, antimicrobial, anticancer

## Abstract

Herbal medicine has been gaining special interest as an alternative choice of treatment for several diseases, being generally accessible, cost-effective and safe, with fewer side-effects compared to chemically synthesized medicines. Over 25% of drugs worldwide are derived from plants, and surveys have shown that, when available, herbal medicine is the preferred choice of treatment. *Origanum syriacum* (Lamiaceae) is a widely used medicinal plant in the Middle East, both as a home and a folk remedy, and in the food and beverage industry. *Origanum syriacum* contains numerous phytochemical compounds, including flavonoids, phenols, essential oils, and many others. Because of its bioactive compounds, *O. syriacum* possesses antioxidant, antimicrobial, and antiparasitic capacities. In addition, it can be beneficial in the treatment of various diseases such as cancer, neurodegenerative disorders, and peptic ulcers. In this review, the chemical compositions of different types of extracts and essential oils from this herb will first be specified. Then, the pharmacological uses of these extracts and essential oils in various contexts and diseases will be discussed, putting emphasis on their efficacy and safety. Finally, the cellular and molecular mechanisms of *O. syriacum* phytochemicals in disease treatment will be described as a basis for further investigation into the plant’s pharmacological role.

## 1. Introduction

Since time immemorial, natural products have been widely used as medications to alleviate and treat disease and illnesses, either as traditional herbal extracts or as pure bioactive compounds. Plants extracts and essential oils abound with secondary metabolites and bioactive components that exhibit a broad spectrum of pathological functions, such as treating colds, diarrhea, indigestion, and inflammation, among others [1,2,3,4]. The isolation of morphine from the opium plant at the start of the nineteenth century marked the beginning of plant-derived medicine or the era of modern drugs [5]. Over the past few decades, plant-based natural products and their derived pharmacologically active compounds have been gaining ever more interest, stimulating herbal medicine research in the drug discovery process. Examples of other commonly used drugs derived from plant sources include salicin from *Salix alba*, the source of the pain killer and blood-thinning medicine aspirin; the cardiac glycoside digoxin from *Digitalis purpurea*; the antimalarial drug quinine from the bark of *Cinchona* species; the antihypertensive agent reserpine from *Rauwolfia serpentina*; anticancer drugs such as taxol isolated from *Taxus* species, and chemotherapeutic agents derived from camptothecin from *Camptotheca acuminata*, to name a few [6,7]. Such a long history and the availability of tens of thousands of plant species make natural compounds undoubtedly a great source of potential drug leads. While chemically synthesizing drugs using modern technology is a tedious and expensive process that is reaching its limits, the development of new drugs from herbal origins should, in theory, have a higher success rate due to the unique structural diversity and complexity of natural compounds associated with a wide range of biological activities [8]. In fact, a large number of drugs today are marketed as derived from or similar to natural herbal compounds [9]. 

## 2. Traditional Uses of *O. syriacum*

*Origanum* is a perennial shrubby plant of the Lamiaceae family, native to the Mediterranean region and widely used in culinary practices, particularly in Lebanon, Syria, Jordan, Egypt, Palestine, and Turkey [10]. The chemical profiling of various *Origanum* species revealed an abundance of bioactive molecules such as flavonoids, glycosides, terpenes, and phenols known to possess various pharmacological properties [11]. For thousands of years, *Origanum* species have been used in traditional medicine to treat various diseases, and recently interest in these plants as important medicinal herbs with promising therapeutic potential has increased greatly. Of particular interest is a native to the Levant region: *Organium syriacum*, also called Syrian oregano, Lebanese oregano, or the hyssop of the Bible. Commonly known as “zaatar”, it is, in fact, a component of the zaatar mixture, which also includes sesame and sumac (*Rhus coriaria* L.). Botanically, *O. syriacum* is an aromatic perennial herb with woody roots and hairy stems, and a height range of 60–90 cm. Like other members of the family, such as *O. compactum* [12] and *O. vulgare* [13], the aerial parts of *O. syriacum* are characterized by the presence of secretory trichomes containing essential oils of important medicinal value, e.g., carvacrol and thymol, giving these plants a unique aroma and flavor [14]. In addition to its use as a culinary herb, its traditional medical uses include relieving stomach and intestinal pain and the treatment of colds, coughs, and toothache [14]. In recent years, *O. syriacum* has been shown to have various biological properties, including antioxidant, anti-inflammatory, anticancer, neuroprotective, antibacterial, and antihelminthic activities [15]. Additionally, toxicity studies have shown that consumption of this plant has no adverse effects in animals, making it an attractive natural agent in drug discovery [16]. This review aims to consolidate the up-to-date phytochemical profile of *O. syriacum* from the literature and provide a comprehensive overview of its pharmacological properties. 

## 3. Methods

Major scientific literature databases, including PubMed, Scopus, Google Scholar, Chemical Abstracts, ScienceDirect, and Medicinal and Aromatic Plants Abstracts, were used to retrieve articles related to the review subject. The search period covered articles published between 1983 and the end of 2021. The search used the keywords and MeSH terms for ‘*Origanum syriacum*’, AND (‘phytochemical content’, ‘pharmacological properties, or activities, or effects, or roles’, ‘anti-inflammatory’, ‘antioxidant’,’ anticancer’, ‘antiparasitic’, ‘antimicrobial’, ‘industry’, ‘neuroprotective’, ‘antimelanogenic’, ‘pharmaceutical formulations’, or ‘antiulcerogenic’).

## 4. Phytochemical Constituents of *Origanum syriacum* Extracts

Although extracts of *O. syriacum* contain numerous phytochemical compounds, few compositional studies have been conducted. Depending on the type of extract, different constituents have been identified. Indeed, several terpenoids or monoterpene glucosides have been isolated from an ethanolic extract of this plant [17]. In addition, this extract is also rich in volatile oils and phenolic compounds, flavonoids being the major constituents, and carotenoids mainly represented by β-carotene [18]. Further investigations showed that the ethanolic extract also contains other phenolic compounds, mainly thymol and carvacrol [19]. On the other hand, the aqueous extract of this plant was shown to be rich in only three compounds: carvacrol, carveol, and thymoquinone [19]. Interestingly, the methanolic extract has novel flavonoids. In fact, a novel prenylated biflavone was isolated [20] as well as a novel flavone glycoside [21]. Furthermore, rosmarinic acid, oleanolic acid, and ursolic acid are also major constituents of the methanolic extract of this plant [22]. Figure 1 and Table 1 summarize the main phytochemical constituents of the different extract types of *O. syriacum*.

## 5. Phytochemical Constituents of *Origanum syriacum* Essential Oils

Essential oils are complex oily volatile liquids synthesized by different aromatic plant organs (flowers, buds, seeds, leaves, twigs, bark, herbs, wood, fruits, and roots) and are stored in secretory cells, cavities, canals, epidermic cells, or glandular trichomes. They are characterized by having a strong odor and are widely used for bactericidal, fungicidal, insecticidal, medicinal, and cosmetic applications, especially in the pharmaceutical, cosmetic, agricultural, and food industries [23]. Interestingly, essential oils from the same plant may differ depending on factors such as rain, soil, time of harvesting, and fertilization [23,24]. 

*Origanum syriacum* essential oils are highly important, and several research teams have worked on their extraction in an attempt to determine their constituents. In most of the studies, the composition of the essential oil was determined by gas chromatography and confirmed by GC/MS analyses. In several recent studies, around 45 constituents were detected; the most prominent ones being thymol, carvacrol, caryophyllene, γ-terpinene, *p*-cymene, terpinene, eugenol, anisaldehyde, sabinene, aromadendrene, spathulenol, t-cadinol, bicyclogermacrene, cadinene, and others [14,24,25]. One of the earliest studies, which was done in 1995 on sample plants collected from a region in Lebanon, showed that the major components of *O. syriacum* are carvacrol and thymol [25]. On the other hand, a study carried out on samples from Palestine showed that terpenes were the prominent constituent [26]. Other studies performed in Egypt showed that *O. syriacum* contains *cis*-sabinene hydrate and terpinen-4-ol, as well as γ-terpinene, thymol, and carvacrol [24,27,28]. In addition, β–myrcene and *p*-cymene were abundant in *O. syriacum* from the Syria region [27]. This confirms the theory that suggests that geographical differences play a role in the composition of the essential oils. Figure 2 and Figure 3 and Table 2 summarize the main findings of the phytochemical constituents of *O. syriacum* essential oil.

## 6. Extraction Methods for *Origanum syriacum*

The essential oil components of *O. syriacum* are not only affected by environmental and geographical factors but also by the extraction method used. Several studies have used hydrodistillation to extract the oils, as this has shown a higher yield than other methods [14,18,24,27,29,31,32,33,34,35]. Hydrodistillation has long been used to extract bioactive materials from plants, and it involves three physiochemical processes: hydro-diffusion, hydrolysis, and decomposition by heat. This method of extraction requires the whole plant, dried and shredded, to be fully immersed in water in a hydrodistillation Clevenger apparatus. The mixture is then boiled for 3 h, and the essential oil is driven with water vapor into a refrigerated tube, then recovered in a glass test tube and purified with anhydrous sodium bicarbonate. Finally, it is stored in the dark at 4 °C in a sealed glass vessel [35]. In an article comparing the extracted oils obtained through conventional hydrodistillation or microwave-assisted hydrodistillation, results showed that microwave-assisted hydrodistillation was three times faster and required less time (around 10 min) to reach the required temperature than the conventional method; moreover, the yield of the former was much greater than that of the latter, and the whole process took less time [36]. In the conventional hydrodistillation method, the plant was placed in a rounded flask that was connected to Clevenger-type apparatus and heated at 100 °C for 3 h using a heat mantle, while in the microwave-assisted hydrodistillation method, a focused microwave apparatus was used, and the plant was extracted using a Clevenger-type apparatus connected outside of the microwave reactor; temperature and power were adjusted to 100 °C and 800 W, respectively. Another study using ultrasound microwave-assisted hydrodistillation extraction yielded different amounts according to the location of the sample plant [26]. In yet another investigation, solid-phase-micro extraction (SPME) and supercritical-fluid extraction (SFE) were used, and the results were compared with those previously obtained through hydrodistillation [37]. The data showed that thymol and carvacrol, which are the two main constituents of the hydrodistilled oil in the majority of the studies, were found in very low concentrations in the emitted aroma of *O. syriacum* using SPME. On the other hand, monoterpene hydrocarbons were identified as the major components, which reflects the actual composition of the volatile substances in the aromatic plants [37]. In addition, carvacrol was found as the major oxygenated terpene by the SFE method, while SPME indicated very low concentrations of the oxygenated monoterpenes [37]. Finally, a study showed that volatile oils are better preserved when using the static headspace (HS) method in comparison to steam distillation (SD) [38]. This further confirms that extraction methods can indeed affect the constituents of the essential oils.

## 7. Pharmacological Activities of *Origanum syriacum*

### 7.1. Antioxidant Activities

Oxidative stress is caused by an imbalance between the production and accumulation of reactive oxygen species (ROS) and the human body’s detoxification ability through antioxidant defenses, resulting in oxidation damage and causing premature aging as well as an increased risk of several diseases such as cancer, inflammatory disorders, diabetes, and neurodegenerative diseases. Oxidation reactions also occur during food deterioration, affecting the storage and quality of food products and their shelf life [39]. In recent years, there has been growing interest in the role of antioxidants in the protection against oxidation damage. Moreover, plants provide a good source of natural antioxidant compounds as safe alternatives to synthetic ones such as butyl hydroxy anisole (BHA) and butylated hydroxytoluene (BHT), which are widely used as preservatives in food and consumer products and have been associated with questionable toxic and carcinogenic effects [39,40,41]. 

The antioxidant capacity of *O. syriacum* has been extensively investigated using a wide range of assays, as summarized in Table 3. Its potent antioxidant effect is attributed to the high content of phenolic components (mostly carvacrol and thymol) present in its extracts; it could also be due to the chemical composition of the plant extracts and the relative proportions and synergy among its constituents, which explains the different antioxidant and biological activities observed [28,31,42,43]. Interestingly, in a recent study comparing the antioxidant capacity of the essential oils of *O. syriacum* from different regions, the authors analyzed the effect of the phytochemical composition of the essential oil, noticing that the higher antioxidant capacity resulting from 2,2-diphenyl-1-picrylhydrazyl (DPPH) inhibition was associated with a high lipophilic content (including the unsaturated terpenes γ-terpinene and α-terpinene). In contrast, the higher the phenolic composition of the essential oil, the lower its antioxidant capacity, suggesting a role for unsaturated hydrocarbons in the scavenging capacity of *O. syriacum* [26]. All told, *O. syriacum* provides an important source of antioxidants that could be used as natural food preservatives with strong antimicrobial activities or in the management and treatment of chronic and oxidative stress-related disorders.

### 7.2. Antimicrobial Activities

Today, antimicrobial resistance to many antibiotics is becoming a major challenge in the treatment of microbial infections and infectious diseases, as well as crop protection and the prevention of food spoilage, threatening the health of humans and societies. It is attributed to evolutionary natural selection and is limiting the effectiveness of many antimicrobial drugs [45]. In order to overcome antimicrobial resistance, researchers have been searching for new agents, with an increasing interest in medicinal plants due to the efficacy of their phytochemical constituents and ethnomedicinal properties [46]. The effectiveness of various extracts of *O. syriacum* against a broad range of pathogenic microorganisms has been extensively studied, including Gram-negative and Gram-positive bacteria and fungi, as described in Table 4. In general, carvacrol-rich content is associated with increased effectiveness against bacteria and fungi. More specifically, in a study using the essential oil of *O. syriacum*, the authors pointed out that an increased lipophilic content was associated with increased antibacterial activity against Gram-negative strains, while a thymol-rich content was more effective against Gram-positive ones [26]. In fact, the antibacterial properties of thymol and carvacrol are linked to cell membrane disruption and the inhibition of ATPase activity, causing the discharge of cellular components and depletion of ATP [33]. 

A wide range of chemical-based disinfectants and cleansers have been recommended for oral hygiene and for the treatment of denture stomatitis; however, these are not without side effects, and many can alter the physical properties of the denture bases, prompting the use of extracts from natural origins as safe and efficient alternatives. The effect of *O. syriacum* on bacteria and fungi commonly present in the oral cavity has been tested in several studies using different types of extract. The essential oil from *O. syriacum* strongly inhibited the growth of *Staphylococcus aureus* and *Streptococcus pneumoniae* bacterial strains, as well as that of *Candida albicans* isolated from the human oral cavity [16]. The antimicrobial activity of the essential oil was further confirmed in the treatment of denture stomatitis, a common mucosal disorder affecting denture wearers and associated with the growth of common oral bacterial and fungal strains, including *Staphylococcus aureus*, *Streptococcus mutans*, and *Candida albicans* [47]. The analysis of the antimicrobial effect of different ecotypes of *O. syriacum* revealed that their effectiveness was attributed to the carvacrol and thymoquinone proportions of their essential oils [47]. In another study evaluating the antimicrobial activity of several medicinal plant extracts against opportunistic infections of the oral cavity, the authors demonstrated that the methanolic extract of *O. syriacum* leaves exhibited the highest antimicrobial activity against *Staphylococcus aureus*, *Pseudomonas aeruginosa*, and *Candida albicans* [4]. These studies attributed this antimicrobial effect to the high concentrations of carvacrol, thymol, and thymoquinone present in the various *O. syriacum* extracts, which work by perforating the bacterial membranes [48]. It is also worth mentioning that oral administration of *O. syriacum* essential oil in rats did not cause any secondary effects as no variations were observed in the routine hematological and biochemical assays [16]. Taken together, these studies further support the use of *O. syriacum* as a major component in antiseptic preparations for general oral hygiene and in preventing oral cavity infections, tooth decay, and denture stomatitis in particular, and as a natural antimicrobial agent used in disinfectants and food preservatives in general. 

### 7.3. Antiparasitic Activities

The methanolic extract and essential oil of *Origanum syriacum* were shown to have a strong anti-parasitic activity against *Acanthamoeba castellanii* cysts and trophozoites [52]. The essential oil was also active against gastrointestinal *Anisakis simplex* larvae [53] in vitro, presenting *O. syriacum* as a potential therapeutic antiparasitic agent.

### 7.4. Pharmaceutical Formulations

Multidrug resistance (MDR) continues to be the major challenge in treating many diseases. Novel strategies to overcome MDR, such as combining drugs with other agents that can oppose the pathogen’s resistance mechanisms, are needed and are, indeed, essential. In fact, a recent study developed a new nano-pharmaceutical formulation derived from natural compounds and biogenic ionic metal cofactors [57]. As such, *O. syriacum* essential oils were combined with Zn (II) Salen complex through co-encapsulation by chitosan nanoparticles [57]. This combination showed enhanced antimicrobial activity when compared to each component alone [57].

### 7.5. Industrial Uses

The safety of synthetic pesticides has come under widespread scrutiny in the past few decades because of their documented harmful effects on humans and the environment, e.g., contamination of soil, water, and other vegetation. Even when safety is not a concern for some pesticides, their effectiveness can become an issue due to the development of pest resistance, causing deleterious effects on crop protection and agricultural productivity, as well as on the control of vector-borne diseases. Pest control research has recently been turning to plant sources, these being considered low in risk and toxicity, with a particular focus on plant-borne essential oils [58,59]. Benelli et al. tested the toxicity of essential oil extracted from *O. syriacum* on two important agricultural pests, the cotton leafworm *Spodoptera littoralis* (Boisduval) and the aphid *Myzus persicae* (Sulzer), as well as on the housefly *Musca domestica* as an insect vector model. Results showed that *O. syriacum* was toxic to these insects while it had no effect on non-target species such as the earthworm *Eisenia fetida*, suggesting that *O. syriacum* could be a good source of active ingredients for botanical insecticides [54]. In another study, *O. syriacum* essential oil was shown to have an insecticidal effect against both the larvae and adults of the mosquito vectors *Culex quinquefasciatus* and *Culex pipens molestus* [53,55]. This effect was attributed to the phenolic monoterpene carvacrol, the main component of the extract, which acts as an acetylcholinesterase (AChE) inhibitor blocking the breakdown of acetylcholine and causing neurotoxic effects in insects.

The use of compounds from plant and herbal sources as natural food preservatives has also been gaining much interest in the food industry. Most of the active ingredients used in grain protectants for example, are unsafe, and many are no longer in use, mandating the need for new green substances to improve insecticidal formulations. To this end, the effect of various plant essential oils as grain protectants was also tested against the adults and larvae of *Prostephanus truncatus* (Horn) and *Trogoderma granarium* Everts; two of the main beetles causing serious damage to stored dry commodities and of major economic importance [56]. In this study, an *O. syriacum* essential oil solution sprayed on wheat and maize was considered an effective *Trogoderma granarium* adulticide and an analysis of the oil’s composition showed it was mainly dominated by carvacrol (82.6%). 

### 7.6. Anticancer Activities

Although great advancements have been made in cancer treatment therapies, cancer is still a global challenge and remains the second leading cause of death worldwide. As conventional treatment regimens are associated with a number of undesired side effects and multidrug resistance, the search for new bioactive compounds from plant-derived sources has been on the rise, empowering modern anticancer therapy [60]. In a study screening different culinary herbs for their cytotoxicity against human breast adenocarcinoma cells in vitro, the authors concluded that an ethanolic extract from the aerial parts of *O. syriacum* exhibited antiproliferative activity against the MCF-7 breast cancer cell line with an IC_50_ value of 6.40 µg/mL, whereas the aqueous extract and essential oil had no cytotoxic activity [32]. This was corroborated in another study also using the ethanolic extract, which showed moderate cytotoxicity on MCF-7 cells, with around 40% of cells dead at 500 µg/mL and an IC_50_ value between 500 and 600 µg/mL [44]. The anti-proliferative and cytotoxic effects of *O. syriacum* were also tested in vitro on human leukemia THP-1 cells [61]. Results showed that the ethanol extract caused a reduction in viability of the THP-1 cells with an IC_50_ value of 2.126 mg/mL and had a significant cytotoxicity effect with an LC_50_ value of 9.646 mg/mL, suggesting that the reduction in cell viability observed is due to its cytotoxic rather than anti-proliferative effect.

While the antimicrobial and antioxidant activities of this plant have been extensively studied, its anticancer efficacy still warrants further investigation.

### 7.7. Anti-Inflammatory Activities

Inflammation is a physiological process by which the immune system becomes activated to defend the body against harmful or foreign stimuli such as injury, infection, and oxidative stress. However, when inflammation becomes chronic, it could trigger several diseases, including cancer, cardiovascular disease, atherosclerosis, diabetes, and obesity as well as autoimmune and neurodegenerative disorders. Indeed, chronic inflammation is now considered the most significant cause of death (more than 50%) worldwide [62]. Anti-inflammatory drugs are among the most commonly prescribed class of drugs in the world [63]. However, the withdrawal of many of these drugs from the market due to health and safety concerns, especially when it comes to treating chronic conditions, is putting the spotlight on plants with anti-inflammatory activities in an effort to identify novel compounds with potential novel therapeutic mechanisms [64]. To this end, Shen et al. carried out a quantitation of the anti-inflammatory constituents in *Oreganum* species and established that a methanolic extract of *O. syriacum* aerial parts had high levels of the triterpenoid acids oleanolic acid and ursolic acid, which are reported to possess potent anti-inflammatory activities [22,65].

In order to monitor the inflammatory response, several biomarkers were used, including, but not limited to, the production of pro-inflammatory cytokines such as interleukin (IL)-6, IL-8, and tumor necrosis factor (TNF)-α, and the anti-inflammatory immune suppression cytokine IL-10. In a study using a methanolic extract of *O. syriacum*, the release of IL-6 was significantly inhibited (more than 80%) while the release of IL-10 was attenuated from human peripheral blood mononuclear cells treated with concanavalin A, a plant lectin that has mitogenic properties able to activate T cells, leading to the activation of the inflammatory response through the release of pro-inflammatory cytokines [4]. This is in accordance with previous studies associating both IL-6 and IL-10 with pathogenesis and inflammation related to microbial infections, as their levels are usually increased during infections [66,67]. 

In another study, the anti-inflammatory potential of both ethanolic and aqueous extracts of the leaves of *O. syriacum* was assessed by testing their effect on oxidative stress-related inflammation. Oxidative stress, mainly via overproduction of reactive oxygen species (ROS) and alteration of the cellular antioxidant status, induces inflammation by altering several processes leading to cell injury and damage, such as lipid peroxidation of membranes, protein oxidation, and DNA damage [68]. These inflammatory stimuli induce the hydrolysis of membrane-bound phospholipids by secretory phospholipases A_2,_ such as sPLA_2_-GV, which release arachidonic acid. The latter is then metabolized to pro-inflammatory mediators (prostaglandins and leukotrienes) by the pro-inflammatory enzymes COX-1 and COX-2 cyclooxygenases and various lipoxygenases such as 5-LOX, which means that inhibitors of these enzymes would be potential therapeutic agents to stabilize membranes and treat inflammation. The authors showed that the ethanolic extract showed significant anti-inflammatory potency, more than the aqueous extract, by enhancing membrane stability in a model using human red blood cells, which show similarity to lysosomal membranes. Additionally, the ethanolic extract strongly inhibited the sPLA_2_-GV enzyme, therefore modulating the generation of arachidonic acid resulting from tissue damage. The extract also showed strong inhibitory potency against COX-1 and 5-LOX enzymes, which was comparable to the anti-inflammatory drug Diclofenac and the lipoxygenase inhibitor Nordihydroguaiaretic acid (NGDA), respectively, therefore inhibiting the accumulation of key inflammatory prostaglandins and leukotrienes through targeting the COX-1 and 5-LOX pathways [19]. A summary of the proposed mechanism of *O. syriacum* action in modulating the inflammatory response is shown in Figure 4.

### 7.8. Neuroprotective Activities

Neurodegenerative disorders such as Alzheimer’s disease and dementia are characterized by a significant reduction in cholinergic neurotransmission, which is mediated by the neurotransmitter acetylcholine at the neuromuscular junction between the motor neurons and skeletal muscles. Symptoms of such disorders can be attenuated by using agents that restore the levels of acetylcholine through the inhibition of acetylcholinesterase enzyme (AChE) [69]. *Origanum syriacum* essential oil has been recommended by herbalists in the treatment of neurological conditions because of its monoterpene-rich composition, which is strongly inhibitory to AChE [70]. Moreover, carvacrol, a main constituent of *O. syriacum* essential oil, was also shown to have a strong AChE-inhibitory property [71]. This was further confirmed in a study by Loizzo et al. monitoring the hydrolysis of acetylthiocholine. Results showed that *O. syriacum* essential oil had a strong inhibitory effect on AChE activity in a dose-dependent manner, emphasizing its benefit as a nutraceutical product in the prevention and treatment of neurological disorders [31].

In another study, *O. syriacum* was shown to play a role in the glutamatergic neurotransmitter system by altering the kinetics of the glutamate receptor, α-amino-3-hydroxy-5-methyl-4-isoxazolepropionic acid receptor (AMPAR), responsible for fast excitatory neurotransmission in the central nervous system [72]. Upon glutamate binding, the receptors are activated, allowing ion influx into the neurons, and producing electrical signals. This state is then rapidly desensitized and deactivated. Ensuring tight regulation of glutamatergic neurotransmission is essential in maintaining important brain functions, and the overactivation of the system is linked to various neurodegenerative and neuropsychiatric disorders such as Alzheimer’s disease, Amyotrophic Lateral Sclerosis (ALS), and epilepsy [73,74]. Here the authors showed that *O. syriacum* essential oil prolonged the desensitized and deactivated phases of the AMPAR subunits by possibly stabilizing the desensitized/deactivated conformation and therefore indirectly reducing their activity and consequently decreasing the risk of neurotoxicity, suggesting that this plant can play a major role in the pathophysiology and treatment of neurological diseases. The neuroprotective activities of *O. syriacum* on fast excitatory neurotransmission in the central nervous system are presented in Figure 5.

### 7.9. Antimelanogenic Activity

Pigmentary changes in the skin are due to abnormal melanin production and accumulation in the epidermis resulting from deregulation of pro-melanogenic factors [75]. Abnormalities in skin pigmentation due to hyperpigmentation can be due to several factors, including hormonal changes, inflammation, age, or sun exposure, leading to disorders of the skin such as melasma, lentigo simplex (freckles), and age spots (solar lentigo). In an effort to find depigmenting agents and skin lighteners, cosmetic-based industries have been focusing on agents that target steps in the melanogenic pathway [76]. The most common active ingredients used for topical applications include hydroquinone, salicylic acid, kojic acid, and niacinamide. However, many are not free from side effects and have been the cause of major health concerns, especially over long-term use, causing a surge in the research for plant-based active ingredients in cosmetics [77,78]. To this end, El-Koury et al. tested the effect in vitro of essential oil from *O. syriacum* on the production of melanin in a model using B16-F1 melanocytes. The results showed that the application of the essential oil reduced the production of melanin by 15% at 40 µg/mL in those cells. In parallel, the melanogenic activity of pure carvacrol, the main component of the *O. syriacum* essential oil, was shown to significantly decrease melanin levels in those cells, implicating carvacrol in the process of melanogenesis [79]. In an effort to investigate the mechanism involved, the effects of the *O. syriacum* essential oil and pure carvacrol were tested on tyrosinase, which is the first enzyme involved in the process of synthesis of melanin and is the rate-limiting step. Results showed that both *O. syriacum* oil and carvacrol inhibited the activity of tyrosinase without affecting the enzyme’s level in B16-F1 melanocytes, suggesting that the extract reduces the level of melanin by blocking melanogenesis. This is in agreement with an earlier study, as *O. syriacum* essential oil at 40 µg/mL and carvacrol at 2.5 µg/mL were able to inhibit the activity of tyrosinase *in tubo* by 80% and 56%, respectively [35]. The proposed mechanism of action of the *O. syriacum* extract involves a competitive inhibition model whereby carvacrol binds to tyrosinase, preventing the binding of its usual substrate tyrosine in a mechanism similar to that involving the commonly used skin-lightening agent hydroquinone in the treatment of hyperpigmentation [76,79] as seen in Figure 6. 

### 7.10. Antiulcerogenic Properties

The formation of peptic ulcers is one of the most common chronic conditions affecting millions of people worldwide. It occurs when open sores develop on the lining of the upper digestive tract. Common treatment regimens involve the use of synthetic drugs that reduce stomach acidity or protect the mucous tissues lining the stomach and upper portion of the small intestine. However, with a high recurrence rate, the need to introduce safe drugs without side effects is growing, with particular interest focused on medicinal plants [80]. Rats with ethanol-induced gastric damage were used to test the anti-ulcerogenic potential of *O. syriacum*, along with other medicinal plants traditionally used in the treatment of gastric ulcers [81]. Results showed that oral administration of the ethanolic extract of *O. syriacum* significantly reduced gastric damage in those rats. In another study investigating the anti-ulcerogenic activity of various plant extracts in prophylactic and curative models, the oral administration of *O. syriacum* ethanolic extract to rats with absolute-ethanol-induced gastric damage was shown to have a similar effect to the anti-ulcer drug Lansoprazole in the prophylactic model only. The ethanolic extract also exhibited high free-radical scavenging activity, further supporting the antioxidant potential of *O. syriacum* and its protective and healing role in ethanol-induced gastric ulceration [82]. 

## 8. Conclusions

*Origanum syriacum* is an important Mediterranean plant, widely used in dietary and culinary practices as well as in traditional phytotherapy. With a particularly rich content of carvacrol and thymol, its major pharmacological effects were associated initially with antimicrobial and pesticidal properties and fighting infections. Recently, this plant has become increasingly interesting as a valuable source of bioactive phytocompounds with antioxidant, anti-inflammatory, and neuroprotective properties. So far, antitumor effects of the plant have also been reported in breast and leukemia cancer cells in vitro, along with anti-melanogenic and anti-ulcerogenic activities. Because of its potential biological activity, *O. syriacum* still has immense therapeutic possibilities that are attracting the attention of many researchers working on the development of novel compounds into safe, effective, and affordable medicines.

## Figures and Tables

**Figure 1 molecules-27-04272-f001:**
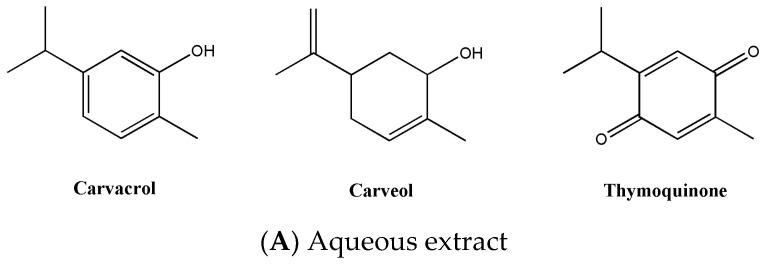
Major phytochemical components of *Origanum Syriacum* extracts. (**A**): Aqueous, (**B**): Ethanolic, and (**C**): Methanolic extracts.

**Figure 2 molecules-27-04272-f002:**
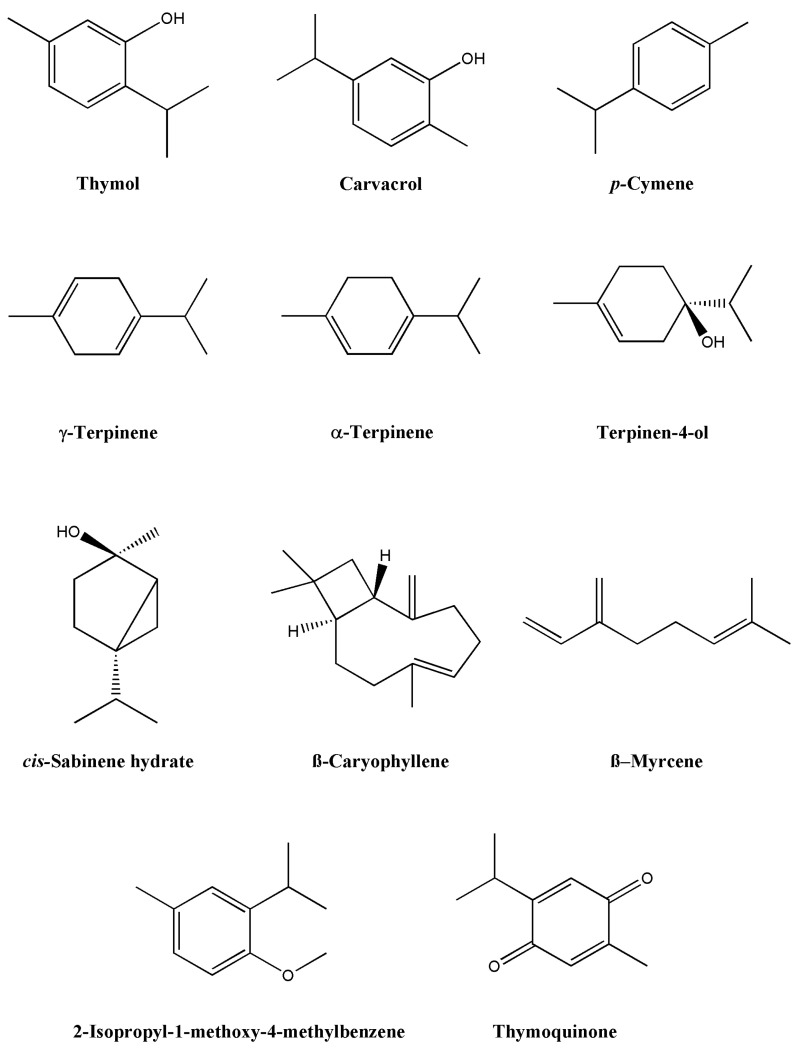
Major phytochemical components of *Origanum syriacum* essential oil.

**Figure 3 molecules-27-04272-f003:**
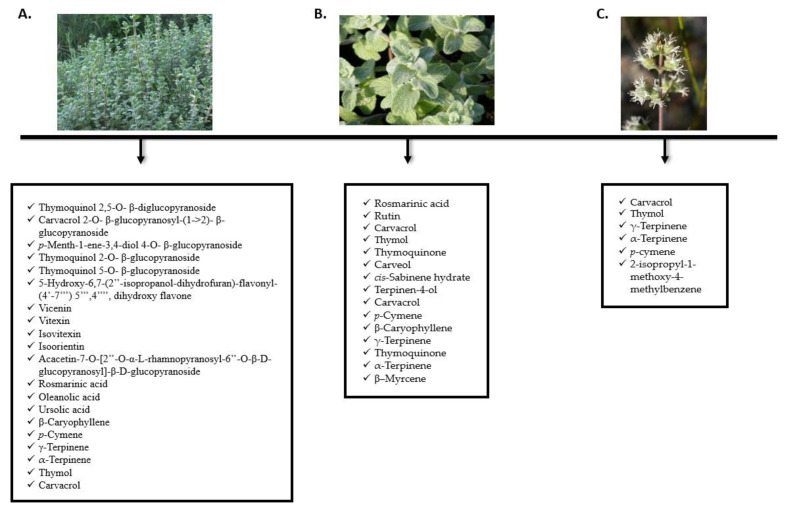
Major metabolites reported from various parts of *Origanum syriacum*. (**A**): Aerial parts, (**B**): Leaves, (**C**): Flowers.

**Figure 4 molecules-27-04272-f004:**
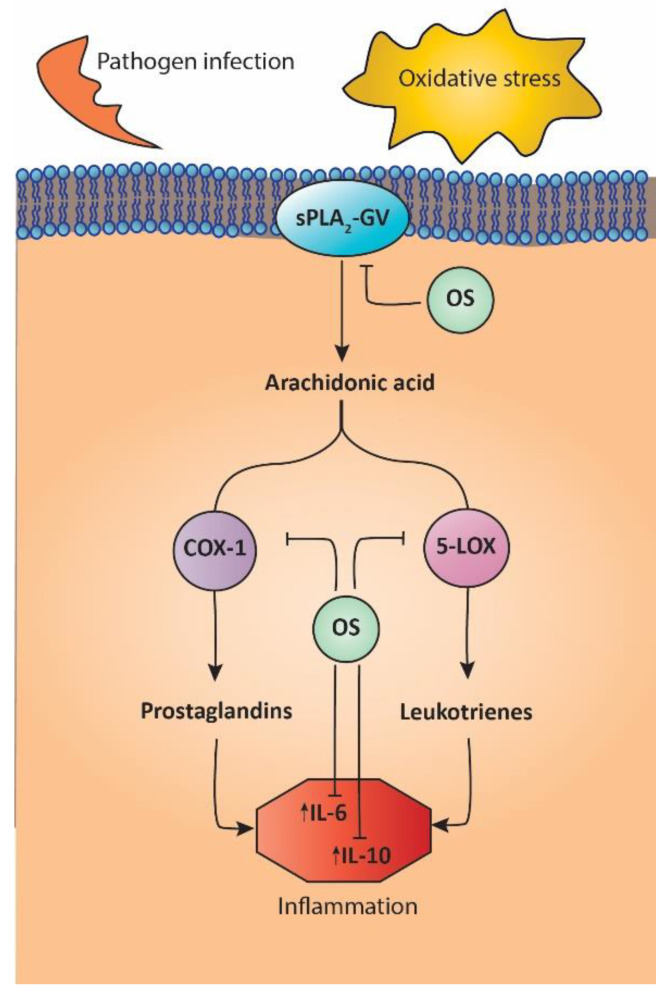
*Origanum syriacum* modulates the inflammatory response. *Origanum syriacum* blocks the hydrolysis of arachidonic acid from phospholipid membranes by the phospholipase A2 sPLA2-GV, which is normally released by inflammatory stimuli resulting from pathogen infection and oxidative stress. *Origanum syriacum* also blocks the conversion of arachidonic acid to pro-inflammatory mediators such as prostaglandins and leukotrienes by inhibiting COX-1 and 5-LOX enzymes, respectively, in addition to inhibiting the production of the pro-inflammatory cytokines IL-6 and IL-10, therefore playing a major role in the pathophysiology of inflammation. OS: *Origanum syriacum*; sPLA2-GV: Secretory phospholipase A2-Group V; COX-1: cyclooxygenase-1; 5-LOX: 5-lipooxygenase; IL-6: interleukin-6; IL-10: interleukin-10.

**Figure 5 molecules-27-04272-f005:**
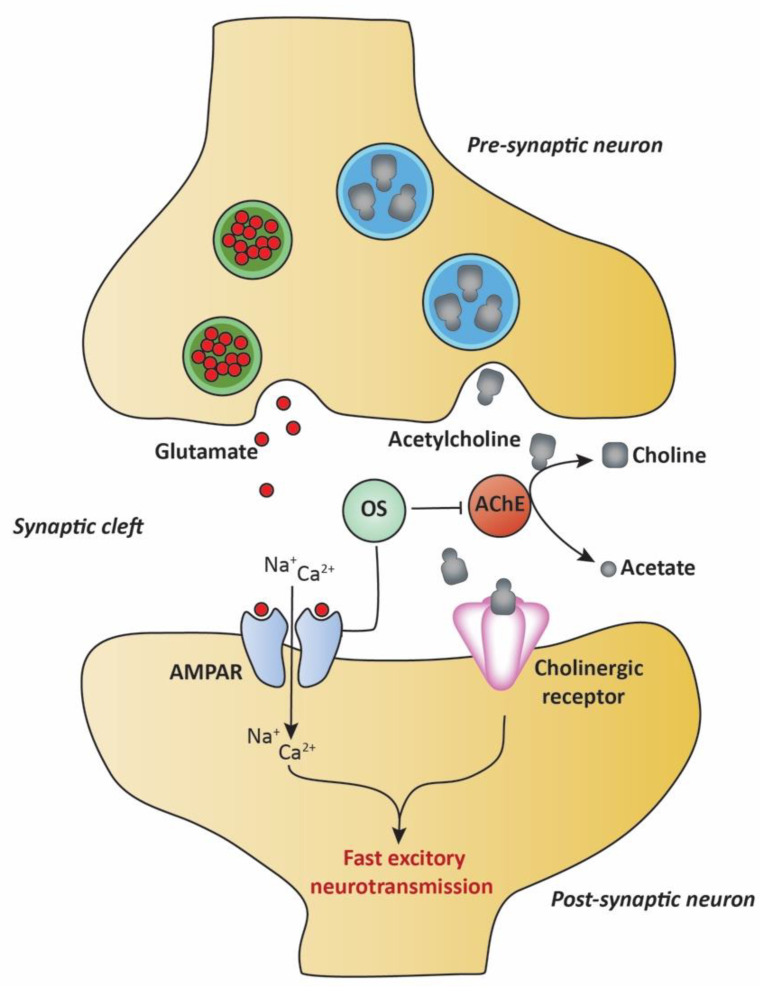
*Origanum syriacum* exhibits neuroprotective activities on fast excitatory neurotransmission in the central nervous system. *Origanum syriacum* regulates the glutamatergic neurotransmission system by stabilizing the desensitized/deactivated conformation of the glutamate receptor AMPAR, thus reducing its activity and decreasing the risk of neurotoxicity. *Origanum syriacum* also restores cholinergic neurotransmission by inhibiting the activity of AChE and preventing the hydrolysis of acetylcholine to choline and acetate. OS: *Origanum syriacum*; AMPAR: α-amino-3-hydroxy-5-methyl-4-isoxazolepropionic acid receptor; AChE: acetylcholinesterase enzyme.

**Figure 6 molecules-27-04272-f006:**
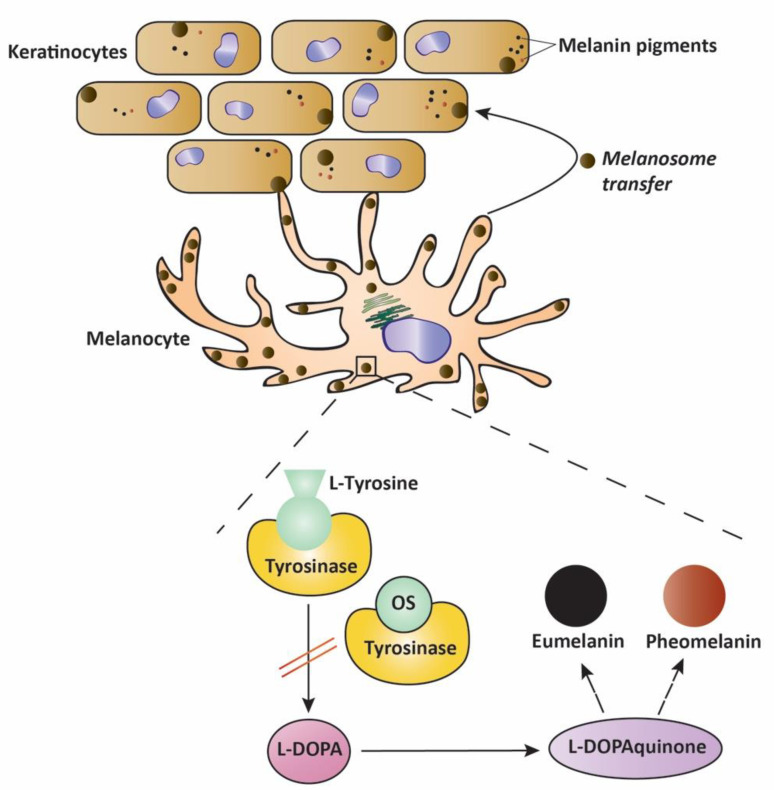
*Origanum syriacum* exhibits antimelanogenic properties by blocking melanogenesis. *Origanum syriacum* reduced the production of melanin by blocking the activity of the tyrosinase enzyme in a competitive inhibition model. OS: *Origanum syriacum*.

**Table 1 molecules-27-04272-t001:** Summary of phytochemical composition of *Origanum Syriacum* extracts.

Plant Part	Extract Type	Technique	Main Results	Ref.
Aerial parts	Ethanolic extract	MS NMR	Isolation of 5 compounds: thymoquinol 2,5-O- β-diglucopyranoside, carvacrol 2-O- β-glucopyranosyl-(1->2)- β-glucopyranoside, *p*-menth-1-ene-3,4-diol 4-O- β-glucopyranoside, thymoquinol 2-O- β-glucopyranoside, and thymoquinol 5-O- β-glucopyranoside	[17]
Aerial parts	Methanolic extract	EI-MS NMR	Identification of 5 compounds: 5-hydroxy-6,7-(2′′-isopropanol-dihydrofuran)-flavonyl-(4′-7′′′) 5′′′,4′′′′, dihydroxy flavone, as well as vicenin, vitexin, isovitexin, and isoorientin	[20]
Aerial parts	Methanolic extract	UV IR NMR	Isolation of 11 flavonoids: acacetin-7-O-[2′′-O-α-L-rhamnopyranosyl-6′′-O-β-D-glucopyranosyl]-β-D-glucopyranoside, luteolin, apigenin, luteolin-6-C-glucoside, luteolin-3′-methylether-6-C-glucoside, luteolin-7,4′ -dimethyether-6-C-glucoside, apigenin-7- methylether-6-C-glucoside, apigenin-7-O-glucoside, diosmetin-7-O-glucoside, acacetin7-O-glucoside, and acacetin-7-O-rutinoside	[21]
Aerial parts	Methanolic extract	LC-MS NMR HPLC	Isolation of rosmarinic acid, oleanolic acid, and ursolic acid	[22]
Leaves	Ethanolic extract	HPLC GC	Isolation of 11 phenolic compounds (catechol was the major one), 9 flavonoids (rutin was the major one) and 24 volatile essential oils (carvacrol and thymol were the major ones)	[18]
Leaves	Ethanolic and Aqueous extracts	GC-MS	12 phytochemicals were isolated in the ethanolic extract with carvacrol and thymol found in high amounts. Only 3 phytochemicals were identified in the aqueous extract (carvacrol, thymoquinone, and carveol)	[19]

**Table 2 molecules-27-04272-t002:** Summary of the essential-oil composition and the volatiles profile from different parts of *Origanum syriacum* plants from different areas.

Plant Part	Technique	Location	Main Results	Ref.
Aerial parts	GC-MS	Egypt	45 constituents were identified; carvacrol is the main component. The carvacrol variety found on Mt. Sinai is noticeably different due to the presence of geraniol, geranyl esters, and ethyl cinnamate	[28]
Leaves and flowers	GLC	Lebanon	Essential oil component was about 3% (wt/vol) and the major components of this oil were carvacrol and thymol	[25]
Herbal parts	GC GC-MS	Egypt	40 constituents were collected; thymol was the main component and interestingly, *cis*-sabinene hydrate and terpinen-4-ol were also major constituents	[29]
Leaves	GC GC-MS	Dortyol district at an altitude of 200 m, Southern Turkey	13 components were characterized. The major components were: γ-terpinene (27.79%), carvacrol (26.97%), *p*-cymene (15.69%), and β-caryophyllene (12.59%)	[30]
Flowering aerial parts	GC GC-MS	Baskinta Mountain at an altitude of up to 1500 m, Lebanon	36 compounds were found, representing 90.6% of the total oil. The most abundant components were thymol (24.7%), carvacrol (17.6%), γ-terpinene (12.6%), *p*-cymene (8.7%), 2-isopropyl-1-methoxy-4-methylbenzene (7.9%), and α-terpinene (2.5%)	[31]
Leaves	GC-MS	Amman	30 constituents were found; carvacol (41.1%), *p*-cymene (30.22%), γ-terpinene (4.27%), and *cis*-sabinene hydrate (3.22%) were the most abundant	[32]
Entire plant (stems, leaves, and flowers)	GC-MS	Sekem company plantation in the city of Bilbeis in the Sharkea region, northeastern Cairo, Egypt	23 compounds were identified, representing 94.5% of the total oil, the major constituents being thymol (21.04%) and γ-terpinene (18.96%)	[33]
Leaves	GC-MS	Arabsalim, South Lebanon	5 constituents were reported; carvacrol (78.4%), thymol (17.9%), and thymoquinone (2.5%) were the most abundant	[14]
Aerial parts	GC	Al-Husn cultivar in Sinai (AlAbtal village), Egypt	23 constituents were found; carvacrol (76.06%), thymol (13.29%), *p*-cymene (1.28%), and γ-terpinene (1.27%) were the most abundant	[18]
Entire plant	GC-MS	Wild plants: Wadi-Firan Cultivated plants: El-Naby Danyal, Saint Katherine in the South Sinai, Egypt	50 constituents were found and differences between constituents of wild and cultivated plants were reported; carvacrol (81.38%) was the most abundant in the cultivated type and thymol (31.73%) in the wild type	[24]
Aerial parts	GC GC-MS	Lebanon	27 constituents were found; carvacol (60.8%), *p*-cymene (8.4%), thymol (7.9%), and γ-terpinene (7.5%) were the major constituents	[34]
Stems and leaves	GC-MS	Lebanon	34 compounds were found; carvacrol (60.1%) was the most abundant	[35]
The branches (leaves and buds)	GC HPLC	Western and coastal governorates of Syria	15 constituents were found; β–myrcene (21.93%), carvacol (19.23%), *p*-cymene (7.67%), and thymol (7.4%) were the most abundant	[27]
Leaves	GC-MS	West Bank, Palestine	17 constituents were found; thymol (19.99%) (39.87%) in Qalqilya and Tulkarm, respectively, and α-terpinene (27.95%) (36.8%) in Jerusalem and Bethlehem, respectively	[26]
Aerial parts	GC GC-MS	South Lebanon	35 constituents were found; α*p*-cymene (4.2–14.8%), γ-terpinene (1.5–10.6%), thymol (37.8–56.3%), carvacrol (10.3–35.8%), and β-caryophyllene (1.2–2.1%) were the most abundant	[23]

**Table 3 molecules-27-04272-t003:** Antioxidant activities of *Origanum syriacum*.

Method of Study	Main Results	Ref.
2,2-diphenyl-1-picrylhydrazyl (DPPH)-radical scavenging activity	The deodorized and non-deodorized methanolic extracts as well as the deodorized hot-water extract had high antioxidant and free-radical scavenging activity with IC_50_ values of 21.40, 26.98 and 42.80 µg/mL, respectively, while the essential oil showed weaker activity with an IC_50_ value of 134.00 µg/mL	[42]
	The essential oil showed a significant scavenging ability in a concentration-dependent manner and had an IC_50_ value of 1.7 μg/mL	[31]
	The water-soluble extract was capable of scavenging DDPH radicals in a concentration-dependent manner with an IC_50_ value of about 400 μg/mL	[43]
	The essential oil showed a strong free-radical-scavenging effect with an IC_50_ value of 6.6 mg/mL	[33]
	The essential oils showed a strong inhibitory effect on DPPH free-radical production in a dose-dependent manner with IC_50_ values ranging from 31.60 to 63.10 μg/mL, depending on the region the plant was taken from in the West Bank area of Palestine	[26]
	The essential oil exhibited a radical-scavenging activity in a concentration-dependent manner, with 17.12% DPPH-radical scavenging at 500 mg/mL	[30]
	The ethanolic extract showed high free-radical scavenging in a concentration-dependent manner, with 50% inhibition at 200 μg/mL and a scavenging capacity IC_50_ of 145 μg/mL	[44]
DPPH by TLC	The essential oil revealed at least three spots with the DPPH reagent: carvacrol, thymol (which had IC_50_ values of 245.00 and 161.70 μg/mL) and *p*-cymene (which showed no antioxidant activity)	[42]
β-carotene-linoleic acid bleaching inhibition	The inhibition of linoleic acid oxidation was observed with all extracts: deodorized hot water (93.1%), deodorized methanol (88.3%) and methanol (81.6%), dichloromethane (80.5%), hexane (71.5%), and essential oil (58.8%), in accordance with the total phenolic content of each extract	[42]
	The essential oil inhibited the oxidation of linoleic acid after 30 min and 60 min incubation with an IC_50_ value of 33.6 μg/mL and 58.9 μg/mL, respectively	[31]
	The ethanolic extract showed high bleaching inhibition at 63.1%	[44]
Ascorbate-iron (III)-catalyzed phospholipid peroxidation	The water-soluble extract showed strong hydroxyl-radical scavenging activity as it was able to inhibit the formation of 2-thiobarbituric acid-reactive species (TBARS) by scavenging hydroxyl radicals generated by ascorbate-iron (III) in a concentration-dependent manner, with an IC_50_ of about 600 μg/mL	[43]
Ferric-reducing antioxidant capacity (FRAC)	The essential oil showed strong ferric-reducing capacity in a concentration-dependent manner with a Trolox concentration of 2.87 mmol/L at 50 g/L	[33]
Iron (II) chelation activity	The water-soluble extract also was shown to be an effective chelator of iron in a dose-dependent manner with approximate IC_50_ value of 0.9 mg/mL	[43]
	The essential oil was capable of chelating iron (II) in a concentration-dependent manner with an EC_50_ value of 0.89 mg/mL, better than ascorbic acid and BHT	[33]
Iron (III) to iron (II) reducing activity	The water-soluble extract was able to reduce iron (III) in a concentration-dependent manner	[43]
Inhibition of lipid peroxidation of buffered egg yolk by the thiobarbituric acid-reactive substance (TBARS) assay	The essential oil showed a strong lipid peroxidation inhibitory ability with an EC_50_ value of 3.99 mg/mL, similar to that of the ascorbic acid control	[33]
Nonsite-specific hydroxyl radical-mediated 2-deoxy-D-ribose degradation	The water-soluble extract was capable of inhibiting TBARS formation with an IC_50_ value of about 1.8 mg/mL by scavenging hydroxyl radicals before reacting with the 2-deoxy-D-ribose substrate	[43]
Site-specific hydroxyl radical-mediated 2-deoxy-D-ribose degradation	The extract was capable of preventing the oxidative degradation of 2-deoxy-D-ribose by interrupting the generation of hydroxyl radicals at an IC_50_ value of about 0.2 mg/mL through iron (III) chelation and deactivation	[43]
Oxidative stability of fat by the Rancimat assay	The essential oil showed some oxidative stability of lard, with an antioxidant activity index (AAI) of 1.1, lower than that of ascorbic acid (1.44) and BHT (2.42)	[33]
Reducing power by the method of Oyaizu	The essential oils showed some reducing power in a dose-dependent manner with an absorbance of 0.77 at 500 mg/mL, lower than that of ascorbic acid (0.96 at 10 mg/mL)	[30]
Thiocyanate method	The essential oil showed an antioxidant activity, as measured by the formation of peroxides, in a concentration-dependent manner that was similar to that of the BHT control	[30]

**Table 4 molecules-27-04272-t004:** Antimicrobial and pesticidal activities of *Origanum syriacum*. MIC: Minimum inhibitory concentration.

Organisms	Used Extract	Main Results	Ref.
**Bacteria**			
*Acinetobacter lwoffi*	Ethanolic	Inhibition zone of 40.00 mm and an MIC of 0.13 mg/mL. In comparison, carvacrol had an MIC of 0.28 mg/mL	[42]
*Bacillus brevis*	Essential oil	Inhibition zone of 37 mm at 4 µg/disc, much higher than the positive controls used, ampicillin (14 mm at 10 µg/disc) and streptomycin (16 mm at 10 µg/disc), attributed to the carvacrol and γ-terpinene in its oil	[30]
*Bacillus cereus*	Essential oil	Inhibition zone of 30.60 mm and an MIC of 0.13 mg/mL. In comparison, carvacrol had an MIC of 0.14 mg/mL	[42]
*Bacillus megaterium*	Essential oil	Inhibition zone of 44 mm at 4 µg/disc, much higher than the positive controls used, ampicillin (11 mm at 10 µg/disc) and streptomycin (17 mm at 10 µg/disc), attributed to the carvacrol and γ-terpinene in its oil	[30]
*Bacillus subtilis*	Essential oil	Inhibition zone of 39 mm at 4 µg/disc, much higher than the positive controls used, ampicillin (15 mm at 10 µg/disc) and streptomycin (18 mm at 10 µg/disc), attributed to the carvacrol and γ-terpinene in its oil	[30]
*Clostridium perfringens*	Essential oil	Inhibition zone of 34.25 mm and an MIC of 0.13 mg/mL. In comparison, carvacrol had an MIC of 0.06 mg/mL	[42]
*Enterobacter aerogenes*	Essential oil	Inhibition zone of 26.50 mm and an MIC of 1.12 mg/mL. In comparison, carvacrol had an MIC of 1.12 mg/mL	[42]
*Enterococcus faecalis*	Essential oil	Inhibition zone of 36 mm at 4 µg/disc, much higher than the positive controls used, ampicillin (16 mm at 10 µg/disc) and streptomycin (17 mm at 10 µg/disc), attributed to the carvacrol and γ-terpinene in its oil	[30]
	Essential oil	Inhibition zone of 19.00 and 11.00 mm for the cultivated and wild-type *O. syriacum* at 4 µL/disc, respectively. In comparison, the inhibition zones of the antibiotics tetracycline, rifampicin, and cefotaxime, were 22.00, 21.00, and 18.00, respectively	[24]
*Escherichia coli*	Essential oil	Inhibition zone of 30 mm at 4 µg/disc, much higher than the positive controls used, ampicillin (11 mm at 10 µg/disc), attributed to the carvacrol and γ-terpinene in its oil	[30]
	Essential oil	Inhibition zone of 26.25 mm and an MIC of 2.25 mg/mL. In comparison, carvacrol and the positive control netilmicin had MICs of 0.56 mg/mL and 1 × 10^−2^ mg/mL, respectively	[42]
	Essential oil	Inhibition zone of 25.00 and 17.00 mm for the cultivated and wild-type *O. syriacum* at 4 µL/disc, respectively. In comparison, the inhibition zones of the antibiotics tetracycline, ceftazidime, rifampicin, cephalexin and cefotaxime were 19.00, 23.00, 15.00, 18.00 and 30.00 mm, respectively	[24]
	Ethanolic	MIC value of 780 µg/mL and an inhibition zone of 14.5 mm using a concentration of 100 mg/mL. In comparison the positive control gentamycin had an inhibition zone of 23.8 mm	[49]
	Essential oil	MIC value of 256 µg/mL	[50]
*Heliobacter pylori*	Ethanolic	Moderate activity against two *Heliobacter pylori* isolates. Inhibition zone of 30 mm with 100 µg/disc. In comparison, the inhibition zones of positive controls used were as follows: ampicillin (55 mm at 10 µg/disc), tetracycline (30 mm at 10 µg/disc)	[51]
*Klebsiella oxytoca*	Essential oil	Inhibition zone of 21 mm at 4 µg/disc, higher than the positive controls used, ampicillin (15 mm at 10 µg/disc) and streptomycin (14 mm at 10 µg/disc), attributed to the carvacrol and γ-terpinene in its oil	[30]
*Klebsiella pneumoniae*	Essential oil	Inhibition zone of 21.75 mm and an MIC of 1.12 mg/mL. In comparison, carvacrol and the positive control netilmicin had MICs of 2.25 mg/mL and 1 × 10^−2^ mg/mL, respectively	[42]
	Ethanolic	MIC value of 390 µg/mL and an inhibition zone of 15.5 mm using a concentration of 100 mg/mL. In comparison the positive control gentamycin had an inhibition zone of 24.8 mm	[49]
*Listeria innocua*	Essential oil	Inhibition zone of 29.00 mm at 100% dose, attributed to thymol, citral, 1,8-cineole, γ-terpinene, *p*-cymene, terpinen-4-ol, and their precursors	[33]
*Moraxella catarrhalis*	Essential oil	Inhibition zone of 24.50 mm and an MIC of 0.13 mg/mL. In comparison, carvacrol had an MIC of 0.28 mg/mL	[42]
*Mycobacterium smegmatis*	Essential oil	Inhibition zone of 24 mm at 4 µg/disc, higher than the positive controls used, ampicillin (19 mm at 10 µg/disc) and streptomycin (15 mm at 10 µg/disc), attributed to the carvacrol and γ -terpinene in its oil	[30]
	Essential oil	Inhibition zone of >60.00 mm and an MIC of 0.54 mg/mL. In comparison, carvacrol had an MIC of 0.14 mg/mL	[42]
*Proteus mirabilis*	Essential oil	Inhibition zone of 26.00 mm and an MIC of 1.12 mg/mL. In comparison, carvacrol had an MIC of 1.12 mg/mL	[42]
*Pseudomonas aeruginosa*	Essential oil	Inhibition zone of 34 mm at 4 µg/disc. In comparison the positive controls used ampicillin (10 mm at 10 µg/disc) and streptomycin (13 mm at 10 µg/disc), attributed to the carvacrol and γ-terpinene in its oil	[30]
	Methanolic	MIC value of 2 mg/mL using a saturated solution of extract in DMSO	[4]
	Essential oil	Inhibition zone of 10.50 mm and an MIC of 9.00 mg/mL. In comparison, carvacrol and the positive control netilmicin had MICs of 4.5 mg/mL and 1 × 10^−2^ mg/mL, respectively	[42]
	Ethanolic	MIC value of 780 µg/mL and an inhibition zone of 14.7 mm using a concentration of 100 mg/mL. In comparison the positive control gentamycin had an inhibition zone of 24.5 mm	[49]
*Pseudomonas vulgaris*	Ethanolic	MIC value of 390 µg/mL and an inhibition zone of 13.8 mm using a concentration of 100 mg/mL. In comparison the positive control gentamycin had an inhibition zone of 23.3 mm	[49]
*Salmonella typhi*	Ethanolic	MIC value of 780 µg/mL and an inhibition zone of 12.8 mm using a concentration of 100 mg/mL. In comparison the positive control gentamycin had an inhibition zone of 23.2 mm	[49]
*Staphylococcus aureus*	Essential oil	Inhibition zone of 36 mm at 4 µg/disc, much higher than the positive controls used, ampicillin (16 mm at 10 µg/disc) and streptomycin (21 mm at 10 µg/disc), attributed to the carvacrol and γ-terpinene in its oil	[30]
	Essential oil	Inhibition zone of 26.50 mm and an MIC of 1.12 mg/mL. In comparison, carvacrol and the positive control netilmicin had MICs of 0.28 mg/mL and 8 × 10^−3^ mg/mL, respectively	[42]
	Essential oil	Inhibition zone of 28–36 mm, depending on the strain used. In comparison, control antibiotics used had inhibition zones ranging from 8 to 32 mm	[16]
	Essential oil	MIC value ranging from 0.781 to 3.132 mg/mL, depending on the ecotype.	[47]
	Essential oil	MIC value of 128 µg/mL	[50]
	Essential oil	Inhibition zone of 32.00 and 24.50 mm for the cultivated and wild-type *O. syriacum* at 4 µL/disc, respectively. In comparison, the inhibition zones of the antibiotics tetracycline, ceftazidime, rifampicin, cephalexin, and cefotaxime were 21.00, 16.00, 32.00, 32.00 and 24.00 mm, respectively	[24]
	Ethanolic	MIC value of 780 µg/mL and an inhibition zone of 15.0 mm using a concentration of 100 mg/mL. In comparison the positive control gentamycin had an inhibition zone of 23.7 mm	[49]
	Methanolic	MIC value of 1 mg/mL using a saturated solution of extract in DMSO	[4]
*Streptococcus mutans*	Essential oil	MIC value ranging from 0.049 to 3.132 mg/mL, depending on the ecotype	[47]
*Streptococcus pneumoniae*	Essential oil	Inhibition zone of 44.00 mm and an MIC of 1.12 mg/mL. In comparison, carvacrol had an MIC of 0.28 mg/mL	[42]
	Essential oil	Inhibition zone of 18–25 mm, depending on the strain used. In comparison, control antibiotics used had inhibition zones ranging from 6 to 20 mm	[16]
*Yersinia enterocolitica*	Essential oil	Inhibition zone of 25 mm at 4 µg/disc, higher than the positive controls used, ampicillin (13 mm at 10 µg/disc) and streptomycin (17 mm at 10 µg/disc), attributed to the carvacrol and γ-terpinene in its oil	[30]
**Fungi**			
*Aspergillus flavus*	Essential oil	MICs of 0.25–2.5 mg/mL and 0.25–5.0 mg/mL for cultivated and wild-type *O. syriacum*, respectively, depending on the strain source. In comparison, the positive control itraconazole had an MIC value of 1.0 mg/mL	[24]
*Aspergillus fumigatus*	Essential oil	MICs of 0.25–2.5 mg/mL and 0.25–5.0 mg/mL for cultivated and wild-type *O. syriacum*, respectively, depending on the strain source. In comparison, the positive control itraconazole had an MIC value of 1.0 mg/mL	[24]
*Aspergillus niger*	Essential oil	MICs of 1.25 mg/mL and 5.0 mg/mL for cultivated and wild-type *O. syriacum*, respectively. In comparison, the positive control itraconazole had an MIC value of 1.0 mg/mL	[24]
	Essential oil	At 0.1 µL/mL, the oil completely inhibited mycelial growth of the fungus. An effect was also observed at 0.02 and 0.05 µL/mL with 22.98 and 54.23% inhibition, respectively	[25]
*Candida albicans*	Essential oil	Inhibition zone of 32–38 mm, depending on the strain used. In comparison, control antibiotics used had an inhibition zone of 6 mm	[16]
	Essential oil	Inhibition zone of >60.00 mm and an MIC of 1.12 mg/mL. In comparison, carvacrol and the positive control Amphotericin B had MICs of 0.28 mg/mL and 1 × 10^−3^ mg/mL, respectively	[42]
	Essential oil	MIC values ranging from 0.024 to 1.563, depending on the ecotype	[47]
	Essential oil	MIC value of 128 µg/mL	[50]
	Ethanolic	MIC values ranging between 150 and 625 µg/mL and inhibition zones of 22.5 and 29.5 mm, respectively, depending on the isolate used and using a crude extract concentration of 100 mg/mL (in comparison the positive control, Amphotericin B had an inhibition zone between 29.5 and 33.5 mm)	[49]
	Methanolic	MIC value of 1 mg/mL using a saturated solution of extract in DMSO	[4]
*Candida krusei*	Essential oil	Strong activity with an inhibition zone of >60.00 mm and an MIC of 1.12 mg/mL (in comparison, carvacrol and the positive control Amphotericin B had MICs of 0.28 mg/mL and 1 × 10^−3^ mg/mL, respectively)	[42]
*Fusarium oxysporum*	Essential oil	At 0.1 µL/mL, the oil completely inhibited mycelial growth of the fungus. An effect was also observed at 0.02 and 0.05 µL/mL with 31.02 and 66.02% inhibition, respectively	[25]
*Microsporum canis*	Ethanolic	84.3% inhibition at a 45 μg/mL, comparable to the positive control Econazole with a 100% inhibition at 5 μg/mL	[49]
*Penicillium species*	Essential oil	At 0.1 µL/mL, the oil completely inhibited mycelial growth of the fungus. An effect of was also observed at 0.02 and 0.05 µL/mL with 41.99 and 93.48% inhibition, respectively	[25]
*Saccharomyces cerevisiae*	Essential oil	Strong activity with an inhibition zone of 28 mm at 4 µg/disc, much higher than the positive controls used, Nystatin (18 mm at 30 µg/disc), attributed to the carvacrol and γ-terpinene in its oil	[30]
*Trichophyton rubrum*	Essential oil	MIC value of 64 µg/mL	[50]
	Ethanolic	93.6% inhibition at a 45 μg/mL, comparable to the positive control Econazole with a 100% inhibition at 5 μg/mL	[49]
**Parasites**			
*Acanthamoeba castellanii*	Methanolic	A concentration of 32 mg/mL killed all trophozoites within 3 h and cysts within 24 h. The effect was concentration dependent.	[52]
*Anisakis simplex*	Essential oil	The oil was active against the larvae with LC_50_ values of 0.087 and 0.067 mg/mL after 24 and 48 h treatment, respectively	[53]
**Pests**			
*Musca domestica*	Essential oil	LC_50_/LD_50_ of 58.7 μg/adult, consistent with that of carvacrol with a value of 59.3 μg/adult	[54]
*Culex quinquefasciatus*	Essential oil	LC_50_ of 36 mg/mL. In comparison carvacrol and thymol had LC_50_ values of 36 mg/L and 37.6 mg/L	[53]
*Culex pipens molestus*	Essential oil	LC_50_ of 36 mg/mL. In comparison, carvacrol and thymol had LC_50_ values of 37.6 and 36 mg/mL, respectively	[55]
*Trogoderma granarium*	Essential oil	Moderate adult mortality at a 500 ppm, with 60% after 7 days of exposure. At 1000 ppm, adult mortality was 93.3% after 4 h, and 97.8% after 6 days of exposure. Moderate larval mortality was observed at both concentrations 7 days post-exposure, with 30.0% and 60% at 500 ppm and 1000 ppm, respectively	[56]
*Myzus persicae*	Essential oil	LC_50_/LD_50_ of 2.1 mg/L. In comparison carvacrol had a value of 1.6 mL/L	[54]
*Spodoptera littoralis*	Essential oil	LC_50_/LD_50_ of 103.3 μg/larva. In comparison that of carvacrol was 38.3 μg/larva	[54]

## Data Availability

Not applicable.

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
