# Peer review of "Origanum syriacum* Phytochemistry and Pharmacological Properties: A Comprehensive Review"

_molecules, 2022, doi:10.3390/molecules27134272_

Round 1
Reviewer 1 Report
The Authors presented an interesting overview of Origanum syriacum Phytochemistry and Pharmacological Properties. The manuscript is based on 81 references. It should be noted that the strongest side of this work is the comprehensive description presented in the tables (I really enjoy this way of presentation - fast track for the most important issue). The presentation of selected chemical structures is beneficial as well. Please check the quality of some descriptions of the compounds - in my pdf some errors occurred.
Some minor editorial improvements:
- Please check the head of table 3.
- What do you mean by "Reducing power by the method of Oyaizu"?
- please add some pictures of the plant (consider making a graphic which will show which part of the plant is in use - not mandatory, due to presented information, but it will improve the quality and perception of the manuscript)
My final recommendation: accept
Author Response
Dear Editor,
We would like to thank the reviewers and you for giving us the chance to submit a revised version of our manuscript.
Please find below our answers to each comments.
Regards
Dr Marc Maresca
Reviewer 1
The Authors presented an interesting overview of Origanum syriacum Phytochemistry and Pharmacological Properties. The manuscript is based on 81 references. It should be noted that the strongest side of this work is the comprehensive description presented in the tables (I really enjoy this way of presentation - fast track for the most important issue). The presentation of selected chemical structures is beneficial as well. Please check the quality of some descriptions of the compounds - in my pdf some errors occurred.
Some minor editorial improvements:
- Please check the head of table 3.
Response: Thank you for pointing this out. The head of table 3 is now adjusted.
- What do you mean by "Reducing power by the method of Oyaizu"?
Response: In order to determine the antioxidant property of the extract, the reducing power of the extract was determined by the method named “Oyaizu”. In brief, different concentrations of the extract were mixed with phosphate buffer and potassium ferricyanide. Then TCA were added, and the mixture is centrifuged. Finally, solution of FeCl3 is added to the supernatant and absorbance is measured. Increased absorbance of the reaction mixture indicated stronger reducing power (indicating an antioxidant potential).
- please add some pictures of the plant (consider making a graphic which will show which part of the plant is in use - not mandatory, due to presented information, but it will improve the quality and perception of the manuscript)
Response: Thank you for this suggestion. Pictures of different parts of the plant is now added in figure 3.
Reviewer 2 Report
The current MS cannot be accepted in its current form. The following issues should be addressed in the Manuscript to be more significant and valuable to the readers.
The plant family name should be added in the abstract and keywords
The plant name should be italicized throughout the whole MS
Remove the pronoun, we, and rephrase all sentences.
English editing is necessary, there are many typing and grammatical mistakes
In the abstract, instead of mentioning herbal medicine and its significance, the authors should include the major constituents and bioactivities.
The introduction lacks sufficient references for the fact that the authors mentioned.
No need to use the plant's full name throughout the whole MS.
The search methodology should be a$dd.
In table 1, it seems like the authors pasted the abstracts of various articles, it should be modified or removed.
A heading about the traditional and ethnomedicinal uses as well as the industrial importance of the plant should be added.
A heading about the reported pharmaceutical formulations and clinical trials should be included.
Authors should follow the journal guidelines in the preparation of their MS.
A graph representing the major metabolites reported from various plant part should be added.
A heading about safety and toxicity studies should be included.
Author Response
Dear Reviewer,
Thank you for your valuable comments and for giving us a chance to improve our manuscript ased on your suggestions.
Please find below our answers.
Reviewer 2 :
The current MS cannot be accepted in its current form. The following issues should be addressed in the Manuscript to be more significant and valuable to the readers.
The plant family name should be added in the abstract and keywords.
Response: Thank you for this point. However, the plant family name is mentioned in the abstract (line 15) and in the keywords (line 25).
The plant name should be italicized throughout the whole MS
Response: Thank you for pointing this out. The plant name is now italicized throughout the whole MS.
Remove the pronoun, we, and rephrase all sentences.
Response: Thank you for this suggestion. The pronoun « we » is removed, and all sentences are rephrased in lines 19-24.
English editing is necessary, there are many typing and grammatical mistakes
Response: Thank you for pointing this out. English editing is done throughout the whole MS.
In the abstract, instead of mentioning herbal medicine and its significance, the authors should include the major constituents and bioactivities.
Response: Thank you for your feedback. The major constituents and bioactivities are now added to the abstract in lines 16-19
The introduction lacks sufficient references for the fact that the authors mentioned.
Response: Thank you for your feedback. More references are added in the introduction.
No need to use the plant's full name throughout the whole MS.
Response: Thank you for your suggestion. Origanum syriacum is now abbreviated into O. syriacum.
The search methodology should be a$dd.
Response: Thank you for your comment. The search methodology is now added in lines 75-78.
In table 1, it seems like the authors pasted the abstracts of various articles, it should be modified or removed.
Response: Agree. Table 1 is now modified.
A heading about the traditional and ethnomedicinal uses as well as the industrial importance of the plant should be added.
Response: Thank you for your suggestion. Traditional uses of Origanum syriacum were covered in the introduction. We change the title into “. Introduction and traditional uses of O. syriacum” (line 27). In addition, the industrial importance of this plant is now mentioned under the title Industrial uses (line 287).
A heading about the reported pharmaceutical formulations and clinical trials should be included.
Response: Thank you for your feedback. It would be interesting to include the pharmaceutical formulations and clinical trials of this plant, but to the best of our knowledge, we could not find any paper with solid information on clinical trials.
Authors should follow the journal guidelines in the preparation of their MS.
Response: Agree. The MS now follows the journal guidelines.
A graph representing the major metabolites reported from various plant part should be added.
Response: Thank you for your comment. A graph representing the major metabolites reported from various plant part is now added in figure 3.
A heading about safety and toxicity studies should be included.
Response: Thank you for your suggestion. However, to our knowledge, there is no clinical evidence to support a specific recommended consumption dose for this plant. Only one study showed that the consumption of this plant’s essential oils had no adverse effects on animals [1] making it poised to drug discovery (lines 71-73).
- Lakis, Z., et al., The antimicrobial activity of Thymus vulgaris and Origanum syriacum essential oils on Staphylococcus aureus, Streptococcus pneumoniae and Candida albicans. Farmacia, 2012. 60.
Round 2
Reviewer 2 Report
Thank you for considering some of my suggestions. However, there are some issues that should be considered.
This section ``Introduction and traditional uses of O. syriacum`` should be divided into two separate sections.
``The methodology for this review involved using PubMed to search for publications….``, authors cannot depend only in one database for getting the data about this plant, especially not all journals articles can appear in PubMed. Other data base as web of Science, Scopus, Google scholar should be checked for this work to be useful for all researchers that can find all data about the plant in one source.
-Similar review has been published in 2021 about the same plant ``Alwafa, R.A.; Mudalal, S.; Mauriello, G. Origanum syriacum L. (Za’atar), from Raw to Go: A Review. Plants 2021, 10, 1001. https:// doi.org/10.3390/plants10051001``. I found that there is no measurable addition of this work from the previously published one.
-A heading about the reported pharmaceutical formulations and clinical trials should be included.
Hassan, Yasser A., Amgad IM Khedr, J. Alkabli, Reda FM Elshaarawy, and Ali M. Nasr. "Co-delivery of imidazolium Zn (II) salen and Origanum Syriacum essential oil by shrimp chitosan nanoparticles for antimicrobial applications." Carbohydrate Polymers 260 (2021): 117834.
-In table 1, it seems like the authors pasted the abstracts of various articles, it should be modified or removed.
Authors just modified one cell in the table.
- English editing is necessary, there are many typing and grammatical mistakes.
- Table 2. Summary of essential oil composition and the volatile profile of Origanum syriacum 149
from different plants parts and from different areas
A paragraph summarize the mentioned information in this table should be given.
Author Response
Reviewer 2, round 2:
Thank you for considering some of my suggestions. However, there are some issues that should be considered.
1- This section ``Introduction and traditional uses of O. syriacum`` should be divided into two separate sections.
Answer: This section has been divided into two.
``The methodology for this review involved using PubMed to search for publications….``, authors cannot depend only in one database for getting the data about this plant, especially not all journals articles can appear in PubMed. Other data base as web of Science, Scopus, Google scholar should be checked for this work to be useful for all researchers that can find all data about the plant in one source.
Answer: Thank you for your feedback. Methodology section for this review has been modified and added as a separate section. The author who had dealt with this point last time was not the one who searched for all the papers for this review. Accordingly, the section was modified to include the full list of databases used.
-Similar review has been published in 2021 about the same plant ``Alwafa, R.A.; Mudalal, S.; Mauriello, G. Origanum syriacum L. (Za’atar), from Raw to Go: A Review. Plants 2021, 10, 1001. https:// doi.org/10.3390/plants10051001``. I found that there is no measurable addition of this work from the previously published one.
Answer: We thank Reviewer 2 for this comment. Although the review mentioned by Reviewer 2 contains numerous information about the plant and its essential oils in term of biological activities, we believe our review contains additional information not found in this review motivating its publication. For example, our review contains information about the bioactive molecules of the plant (not described in Alwafa et al) and information about the effect of plant parts and extractions methods. In addition, although the review of Alwafa contains some information about antimicrobial and antioxidant effects of the plant, in term of therapeutic activities, our review presents antimicrobial and antioxidant activities but also other activities such as anticancer, pesticidal effects, antiparasitic, anti-inflammatory, neurologic, anti-melanogenic, antiulcerogenic effects not detailed in Alwafa et al.
-A heading about the reported pharmaceutical formulations and clinical trials should be included.
Hassan, Yasser A., Amgad IM Khedr, J. Alkabli, Reda FM Elshaarawy, and Ali M. Nasr. "Co-delivery of imidazolium Zn (II) salen and Origanum Syriacum essential oil by shrimp chitosan nanoparticles for antimicrobial applications." Carbohydrate Polymers 260 (2021): 117834.
Answer: Thank you for your suggestion. A heading about pharmaceutical formulation has been added in lines 290-298.
-In table 1, it seems like the authors pasted the abstracts of various articles, it should be modified or removed.
Authors just modified one cell in the table.
Answer: Table one is now modified further.
- English editing is necessary, there are many typing and grammatical mistakes.
Answer: Thank you for comment. The manuscript was read and edited by Dr. Sandra Hillmen, a British professor and former editor of Planta, who is acknowledged in the manuscript.
- Table 2. Summary of essential oil composition and the volatile profile of Origanum syriacum
from different plants parts and from different areas
A paragraph summarize the mentioned information in this table should be given.
Answer: Thank you for your suggestion. A paragraph summarizing table 2 has been modified in lines 142-149.